# Long Non-Coding RNAs in Biliary Tract Cancer—An Up-to-Date Review

**DOI:** 10.3390/jcm9041200

**Published:** 2020-04-22

**Authors:** Dino Bekric, Daniel Neureiter, Markus Ritter, Martin Jakab, Martin Gaisberger, Martin Pichler, Tobias Kiesslich, Christian Mayr

**Affiliations:** 1Institute of Physiology and Pathophysiology, Paracelsus Medical University, 5020 Salzburg, Austria; Dino.Bekric@pmu.ac.at (D.B.); Markus.Ritter@pmu.ac.at (M.R.); martin.jakab@pmu.ac.at (M.J.); martin.gaisberger@pmu.ac.at (M.G.); t.kiesslich@salk.at (T.K.); 2Institute of Pathology, Paracelsus Medical University/Salzburger Landeskliniken (SALK), 5020 Salzburg, Austria; d.neureiter@salk.at; 3Cancer Cluster Salzburg, 5020 Salzburg, Austria; 4Ludwig Boltzmann Institute for Arthritis and Rehabilitation, Paracelsus Medical University, 5020 Salzburg, Austria; 5Gastein Research Institute, Paracelsus Medical University, 5020 Salzburg, Austria; 6Research Unit of Non-Coding RNAs and Genome Editing, Division of Clinical Oncology, Department of Medicine, Comprehensive Cancer Center Graz, Medical University of Graz, 8036 Graz, Austria; martin.pichler@medunigraz.at; 7Department of Internal Medicine I, Paracelsus Medical University/Salzburger Landeskliniken (SALK), 5020 Salzburg, Austria

**Keywords:** biliary tract cancer, cholangiocarcinoma, gallbladder cancer, long-noncoding RNA, lncRNA, EZH2

## Abstract

The term long non-coding RNA (lncRNA) describes non protein-coding transcripts with a length greater than 200 base pairs. The ongoing discovery, characterization and functional categorization of lncRNAs has led to a better understanding of the involvement of lncRNAs in diverse biological and pathological processes including cancer. Aberrant expression of specific lncRNA species was demonstrated in various cancer types and associated with unfavorable clinical characteristics. Recent studies suggest that lncRNAs are also involved in the development and progression of biliary tract cancer, a rare disease with high mortality and limited therapeutic options. In this review, we summarize current findings regarding the manifold roles of lncRNAs in biliary tract cancer and give an overview of the clinical and molecular consequences of aberrant lncRNA expression as well as of underlying regulatory functions of selected lncRNA species in the context of biliary tract cancer.

## 1. Introduction

### 1.1. Biliary Tract Cancer

Biliary tract cancer (BTC) is estimated to be the fifth most common gastrointestinal malignancy with a very poor five-year survival rate [1]. Based on the anatomic location of the tumor, BTC can be classified as intrahepatic cholangiocarcinoma (IHC), extrahepatic cholangiocarcinoma (EHC), which can be further subdivided into distal and perihilar subtypes, and gallbladder cancer (GBC) [2,3]. On the histopathological level, IHC and EHC can be classified as tumors with mucin-producing glands or non-mucin-producing glands or, most commonly, as a mixed type [2,3]. Besides these anatomic and histopathological classifications, BTC also shows alterations on a genomic level. For example, large-duct IHCs show a high mutation frequency of classical tumor suppressor genes KRAS and p53, whereas for small-duct IHCs, mutations in IDH1/2 genes are observed [4]. BTC shows a strong geographic variation: in the Western World, incidences are estimated with 0.5–2 per 100,000 population per year, whereas in some regions of South-East Asia incidences are as high as 60 per 100,000 population per year [5,6]. One of the main reasons that causes this discrepancy is the common intake of raw and undercooked food in South-East Asia and the subsequent infection with liver flukes, whereas in the Western Word, major risk factors include primary sclerosing cholangitis, infection with hepatitis B and C viruses and excessive alcohol consumption [6,7,8,9,10]. One major problem in clinical BTC management is that the symptoms are rather unspecific (painless jaundice, abdominal pain, unspecific weight loss) [11]. Therefore, in most cases, BTC is often diagnosed at an already advanced stage, eliminating the option of putative curative surgery [12]. The common treatment for advanced BTC includes palliative treatment, radiation therapy and chemotherapeutic treatment using a combination regimen of cisplatin and gemcitabine, resulting in a median survival of only about one year [12,13,14]. Second line therapeutic options are not standardized and only show weak or moderate improvement in survival rates [15]. The lack of efficient therapeutic options and the high mortality rates demand for a better understanding of the underlying molecular mechanisms of BTC development and progression [16]. Recent studies suggest that long non-coding RNAs (lncRNAs) might be essentially involved in tumorigenesis via different molecular mechanisms [17].

### 1.2. Long Non-Coding RNAs

Once considered as genomic “junk”, it is now clear that DNA regions coding for non-coding RNA species are essential for cellular functions [18]. Non-coding RNAs can be distinguished by their size and are classified as small non-coding RNAs (e.g., sno-, sn-, piwi-, micro-RNAs) and lncRNAs with a size between 200 bp and 100 kb [19]. LncRNAs were first discovered in a study about X-chromosome inactivation about 30 years ago [20]. In this study, the lncRNA XIST was found to be involved in the inactivation of the second X-chromosome [20]. Since then, numerous studies revealed that lncRNAs play a crucial role in several biological processes such as embryogenesis, epigenetic gene expression regulation, cell structure integrity, cell cycle, heat shock response and stem cell pluripotency [21,22,23,24,25,26,27].

In comparison to protein-coding loci, genomic regions coding for lncRNAs show similar epigenetic regulation such as histone modifications, post-transcriptional processing (5′-capping and 3′-polyadenylation) and alternative splicing [19,28]. Moreover, lncRNAs are expressed in a tissue-specific manner. Their median half-life is approximately 3.5 h (mRNA: 7.1 h) and their expression levels in general are relatively low [19,28,29,30,31]. Following the NONCODE database, about 100,000 lncRNA genes are estimated to exist in the human genome, although this number might be an underestimation [32].

LncRNA synthesis is executed by the RNA polymerases II and III and the single-polypeptide nuclear RNA polymerase IV [33,34,35]. Based on the genomic loci from which lncRNAs are transcribed, they can be categorized into six subtypes: long intergenic lncRNAs, intronic lncRNAs, sense lncRNAs, natural antisense transcripts and bidirectional and miscellaneous lncRNAs. Most described lncRNAs are involved in the regulation of protein-coding gene expression [19]. They can regulate their target genes in a cis-regulatory way, such as XIST and ANRIL or in a trans-regulatory way, such as GAS5, FLANC and HOTAIR [29,36]. However, it becomes more and more clear that the specific mechanisms through which lncRNAs directly or indirectly regulate gene expression are manifold, which subsequently allows functional-based categorization of lncRNAs in five major classes [19].

First, lncRNAs can function as molecular signals to recruit transcriptional machinery proteins to target gene promoters. For example, MEG3, which is described as a tumor-suppressive lncRNA, can interact with transcription factors such as RNAPII or p53, change the chromatin structure and thereby promote transcription or inhibition of target genes (e.g., the growth differentiation factor 15 (GDF15)) [19,37]. Second, lncRNAs can function as “guidance-lncRNAs”, as they are able to interact with several chromatin-modifying enzymes, such as histone methylases (e.g., EZH2, G9a), histone acetylases (e.g., MOZ/MORF) or histone deacetylases (HDAC), to guide these epigenetic regulators to their promoter regions for epigenetic regulation of gene expression [19,38]. The importance of this specific function of lncRNAs is underlined by the fact that some epigenetic regulators (e.g., G9a) do not possess DNA binding motives themselves and are therefore dependent on interaction partners such as lncRNAs to be able to find their target genomic region [19,34,38,39]. For example, the lncRNA ANRIL is able to recruit EZH2 to the promoter region of KLF2, a known tumor suppressor, to induce its epigenetic silencing by trimethylation of the histone H3 at lysine 27 (H3K27me3) [40]. Third, lncRNAs may serve as a platform (‘scaffold lncRNAs’) to allow the assembly of multi-component complexes, in which several molecular factors and regulators are assembled at different domains of the scaffold lncRNA [19]. Importantly, this assembly is the prerequisite for the correct regulatory function of the complex [19]. For example, the lncRNA HOTAIR can simultaneously bind the EZH2 along with LSD1, REST and CoREST to perform H3K27me3 as well as demethylation of H3K4, causing transcriptional silencing of target genes [19,41,42,43]. The fourth category consists of “decoy lncRNAs” which are able to titrate specific proteins which themselves are transcriptional or epigenetic regulators [19]. By doing this, they act as molecular sinks, as they only “trap” their targets without altering their functions [19,23]. The decoy function of lncRNAs therefore represent an important regulatory mechanism for gene expression control [19]. The lncRNA GAS5 for example mimics the glucocorticoid response element (GRE) of glucocorticoid receptors (GR) by binding to its DNA binding domain to compete with target DNA GREs for the GR binding. By this, they suppress the transcriptional activity of certain steroid hormone receptors related to glucocorticoid metabolism [23]. The fifth functional lncRNA archetype is termed “competing endogenous RNA” (ceRNA) and regulates gene expression on the post-transcriptional level by sponging specific micro-RNA (miR) species [19,44]. Without the presence of specific ceRNAs, such as pseudogenes, lncRNAs and circular RNAs, miRs can inhibit target mRNA translation in a RISC-dependent manner, via the respective miR response elements [45,46,47,48]. CeRNAs possess the same miR response elements as the target mRNAs and are thereby able to “sponge” miRs which results in translation of the specific mRNA species [45,46,47,48]. This interaction adds a new layer of complexity to post-transcriptional gene expression regulation, as ceRNAs and miRs form a complex network to regulate gene expression on a transcriptome level. Moreover, there is evidence that ceRNAs might possess multiple miR response elements and are therefore able to interact with several miR species, which further demonstrates the importance of ceRNA regulatory networks in regulating gene expression [49,50]. For instance, the lncRNA ANRIL can sponge miR-144, which leads to the translation of the miR-144 target PBX3, a speculated PI3K/AKT and JAK/STAT pathway activating protein, which in turn results in enhanced tumorigenesis [46]. In BTC, Wang et al. could construct a ceRNA-miRNA-mRNA network, consisting of more than 206 RNA species and 450 interactions, pointing out the complexity and importance of ceRNA-based regulatory networks also in BTC [51].

This general categorization of lncRNAs not only demonstrates their functional diversity, but also underlines their central involvement in the regulation of various cellular processes, specifically the direct and indirect regulation of gene expression [30]. Unsurprisingly, numerous studies found that aberrant lncRNA expression and functionality is involved in cancer development and progression and is associated with diverse oncogenic pathways and mechanisms [29,52]. Mounting evidence suggests that lncRNAs also play a role in BTC. For future management of these patients, this is especially interesting considering the fact that a better understanding of the underlying molecular mechanisms of BTC genesis and progression is urgently required in order to develop new therapeutic strategies [17,29].

## 2. LncRNAs in Biliary Tract Cancer

During the last few years, more and more studies have found a connection between BTC and lncRNAs [17]. As summarized in Figure 1A and Table 1, most lncRNAs described in BTC might have an oncogenic role, as they are overexpressed in BTC patient specimens compared to non-tumor tissue and their enhanced expression correlates with unfavorable clinical characteristics (see Figure 2). However, some lncRNAs show a lower expression in BTC samples compared to surrounding healthy tissue, which is associated with poor prognosis, suggesting that these lncRNA species possess tumor-suppressive abilities (Figure 1B).

Consequently, using different in vitro models, knockdown of oncogenic lncRNAs generally resulted in reduced BTC cell proliferation and viability, diminished migration, invasion and epithelial-to-mesenchymal transition (EMT), reduced cancer stem cell (CSC) traits as well as induction of apoptosis and increased chemosensitivity. On the contrary, artificial overexpression of suppressive lncRNAs resulted in a reduction of oncogenic characteristics of BTC cells. As listed in Table 2, lncRNAs promote BTC tumorigenesis via several mechanisms: ceRNA, guide-lncRNA, scaffold-RNA as well as other specific interactions. However, most of the oncogenic lncRNAs in BTC act as ceRNAs, i.e., they positively regulate oncogenic pathways and mechanisms via sponging tumor-suppressive miRs.

Another commonly observed regulatory mechanism of lncRNAs in BTC includes their direct interaction with (epigenetic) regulators and their subsequent guidance to target gene loci. As described below, most guide-lncRNAs currently identified in BTC interact with EZH2, the enzymatic subunit of the epigenetic regulator PRC2, and thereby guide the PRC2 and EZH2 to their target genes for epigenetic silencing [108].

Moreover, lncRNAs also act as scaffolds in BTC cells as demonstrated for the lncRNA SPRY4-IT1, which enables the formation of a functional complex consisting of the three epigenetic regulators DNMT1, EZH2 and LSD1 [97]. It is also worth mentioning that individual lncRNA species might act via different molecular mechanisms as SPRY4-IT1 not only serves as a scaffold, but also possesses the ability to sponge miR-101-3p, which exemplifies the complexity of lncRNA-based regulatory mechanisms [97]. Interestingly, although most lncRNAs in BTC were identified as oncogenic lncRNAs, the few described tumor-suppressive lncRNAs also cover the spectrum of the lncRNA-specific molecular mechanisms, meaning that tumor-suppressive ceRNAs, scaffold lncRNAs and guidance-lncRNAs were identified (Table 2 and Figure 1B). In the following paragraph, we will provide a detailed insight of the underlying molecular mechanisms of individual lncRNAs that are involved in BTC pathogenesis.

## 3. DILC

The cancer stem cell (CSC) model suggests that tumors are a heterogeneous cell containing a subpopulation of cells with stem cell characteristics termed CSCs [109]. Due to their specific characteristics, CSCs are believed to have high clinical relevance as they are associated with tumor aggression, tumorigenicity, formation of metastases and chemoresistance [109]. Typical characteristics of CSC include altered metabolism, expression of specific surface markers, a specific epigenetic profile, enhanced DNA repair activity as well as their dependency on stem cell and pluripotency pathways such as the JAK/STAT3, TGF-β and Wnt/β-catenin signaling cascades [110,111,112,113,114]. Currently, two CSC models are discussed. Following the “hierarchical model”, only a specific predetermined fraction of cells within the tumor possess CSC characteristics, whereas the “stochastic model” postulates that every tumor cell can become a CSC dependent on random events such as mutations, epigenetic alterations and changes in the cellular environment [115]. There is evidence that CSCs are also involved in BTC and that targeting CSCs in BTC might be a promising therapeutic strategy, although the origin of CSCs in BTC is still under debate [116,117,118,119,120,121,122,123,124]. Moreover, several studies connected the expression of classical CSC markers to enhanced tumorigenicity in BTC. Wang et al. demonstrated that BTC cells expressing the CSC markers CD24, CD44 and EpCAM possess high oncogenic potential in vivo compared to BTC cells lacking these protein markers [125]. Furthermore, studies could also link increased expression of pluripotency markers NANOG, OCT4 and CD133+ with a higher resistance towards chemotherapeutics in BTC cells [126]. There is a growing body of evidence that lncRNAs are involved in CSC regulation [127,128]. One lncRNA related to CSCs is DILC which was first described as a tumor-suppressive lncRNA in hepatocellular carcinoma due to its ability to inhibit the IL-6/STAT3 pathway [111]. Moreover, DILC was found to be downregulated in colorectal cancer, leading to suppressed cell proliferation and metastasis [129].

Regarding a potential connection between DILC and BTC, Liang et al. measured expression levels of DILC in GBC tissues compared to normal gallbladder specimens and found enhanced expression of DILC in tumor samples [62]. CSCs can be identified via specific surface and/or functional characteristics [126,130,131]. Although there is no defined set of CSC surface markers in BTC, CD24, CD44, CD133 as well as EpCAM are frequently associated with CSC features in BTC [132]. Interestingly, Liang et al. demonstrated high levels of DILC specifically in the CD44+ and CD133+ subpopulation [62]. They also demonstrated a central role of DILC regarding the maintenance of the CSC status in GBC cells, as RNA interference-based knockdown of DILC not only diminished the number of CD44+ and CD133+ cells, but also resulted in significantly decreased tumorigenicity both in vitro and in vivo [62]. Furthermore, knockdown of DILC also diminished migration of GBC cells [62].

CSCs often rely on pluripotency pathways such as the Wnt/β-catenin pathway [133]. Knockdown of DILC in BTC cells resulted in reduced levels of β-catenin and reduction of CSC characteristics in the same magnitude as the specific Wnt/β-catenin inhibitor FH535, suggesting that DILC is essential for CSC maintenance via activation of the Wnt/β-catenin pathway [62].

## 4. EPIC1

EPIC1 is described as the most frequently epigenetically activated and therefore overexpressed lncRNA in cancer [134]. In most investigated cell lines and tumor samples from various tumor entities, EPIC1 was overexpressed and correlated with poor prognosis and enhanced tumor cell proliferation [134,135,136].

Li et al. confirmed the upregulation of EPIC1 also in BTC tumor tissue samples [63]. To investigate the molecular oncogenic effects of EPIC1 in BTC, RNA interference experiments were performed and showed that knockdown of EPIC1 resulted in apoptosis and suppression of BTC cell growth as well as colony formation [63]. On the contrary, overexpression of EPIC1 promoted cell growth, colony formation and inhibited apoptosis in vitro [63]. It was also found that knockdown of EPIC1 results in downregulation of MYC target genes cyclin A, cyclin D and cdk4 [63,137,138,139]. To further investigate the potential MYC-dependent oncogenic role of EPIC1 in BTC cells, Li et al. performed RNA immunoprecipitation and found a direct interaction between MYC and EPIC1 [63]. Furthermore, they demonstrated the specific dependency of the oncogenic effects of EPIC1 in BTC cells on MYC, as overexpression of EPIC1 in MYC-deficient cells did not lead to cellular oncogenic effects such as enhanced cell growth or colony formation [63].

## 5. CCAT 1

As described earlier, one major mechanism of lncRNAs is to function as ceRNAs and thereby to regulate the transcription of several target genes by sponging specific miR species [19]. CCAT1 was first described in colon cancer where its upregulation was associated with advanced TNM-stage, lymph node invasion, histological differentiation and poor overall survival [140]. Today, studies describe an involvement of CCAT1 in several cancer entities, including BTC [59,60,141,142,143,144,145]. Two studies demonstrated an oncogenic role of CCAT1 via its ability to sponge tumor-suppressive miRs in BTC [59,60].

Micro-RNA-152 is described as a tumor-suppressive miR in several tumor entities [146]. Zhang et al. showed in BTC cells that CCAT1 promotes EMT and invasion in an miR-152-dependent manner, i.e., CCAT1 is able to sponge miR-152 [59]. Knockdown of CCAT1 resulted in enhanced levels of miR-152 as well as of the epithelial marker E-Cadherin, accompanied by decreased expression of mesenchymal markers Vimentin and N-Cadherin, and, consequently, suppression of EMT and invasion [59]. The ability of CCAT1 to promote invasiveness in BTC via sponging specific miR species was also demonstrated in another study: Ma et al. showed that CCAT-1 sponges miR-218-5p, which is a negative regulator of cell proliferation and migration [60,147,148]. Interestingly, knockdown of CCAT1 not only led to suppressed proliferation and invasiveness of BTC cells, but also resulted in elevated protein levels of the miR-218-5p target BMI-1 [60]. BMI-1 is a core component of the PRC1, an epigenetic regulator that is aberrantly active in various cancer types including BTC [145]. Interestingly, BMI-1, mRNA and CCAT1 possess the same miR-218-5p response element. Consequently, enhanced levels of CCAT1 resulted in more effective sponging of miR-218-5p and, ultimately, in elevated BMI-1 protein expression [60].

Although the current literature suggests that CCAT1 promotes BTC tumorigenesis via its ability to sponge tumor suppressive miRs, it is likely that CCAT1 can also act as an oncogenic lncRNA by other mechanisms, as already demonstrated in other cancer entities (i.e., as a scaffold for epigenetic regulators [149]). Therefore, future studies should also focus on the investigation of the non-ceRNA-oncogenic role of CCAT1 in BTC.

## 6. LINC00152

The lncRNA LINC00152 was first described in hepatocellular carcinoma [150]. Since then, studies have showed upregulation of LINC00152 in various tumor types such as gastric and colon cancer [151,152]. Recently, two reports suggest an oncogenic role of LINC00152 in BTC [75,76]: expression of LINC00152 was elevated in BTC and correlated with advanced TNM-stage and lymph node invasion [76]. The underlying oncogenic mechanisms of LINC00152 were investigated using RNA interference: treatment of BTC cells with siRNA against LINC00152 resulted in reduced cell proliferation, migration and invasiveness, respectively, as well as induction of apoptosis [76]. Interestingly, Cai et al. identified SP-1 as a transcription factor that regulates LINC00152 expression as overexpression of SP-1 resulted in enhanced levels of LINC00152 and vice versa [76]. Furthermore, it was shown that siRNA-based repression of LINC00152 resulted in lower levels of p-PI3K and p-AKT, suggesting that LINC00152 might be a positive regulator of the oncogenic PI3K/AKT pathway [76,153]. Since SP-1 is a known oncogenic factor [154,155,156], the authors suggested a model regarding LINC00152 in BTC in which upregulation of SP-1 causes enhanced LINC00152 levels and, consequently, PI3K/AKT pathway activity [76].

Besides its ability to promote BTC via regulation of the PI3K/AKT pathway, LINC00152 can also act as a ceRNA [75]. Using miR pull-down assays, the authors identified miR-138 as a direct interaction partner of LINC00152 [75]. Furthermore, they showed that overexpression of LINC00152 resulted in enhanced HIF-1α (a known miR-138 target, [157,158,159]) and that this effect was reversible via artificial expression of miR-138, suggesting that LINC00152 might positively regulate HIF-1α by sponging miR-138 in BTC [75]. Additionally, by sponging the anti-EMT miR-138, LINC00152 also promotes EMT and migration and invasion of BTC cells [75].

## 7. PAGBC

PAGBC (also known as Long Intergenic Non-Coding RNA 1133 (LINC01133)) has a dual role regarding tumorigenesis. In non-small-lung cancer, studies show that PAGBC promotes tumor cell proliferation via its ability to serve as a scaffold lncRNA for EZH2 and LSD1 [160], whereas in colorectal cancer, PAGBC acts as a tumor-suppressive lncRNA through direct binding to SRSF6, a serine-arginine-rich splicing factor that is involved in various RNA associated processes. [161,162]. Moreover, PAGBC is also described as an oncogenic ceRNA that sponges miR-422a [87,163].

Wu et al. profiled the expression of lncRNAs in GBC compared to non-tumor tissue [87]. PAGBC was identified as an lncRNA that was overexpressed in GBC and, based on additional gene-coexpression network analysis, appears to possess significant oncogenic potential in GBC [87]. To evaluate the clinical importance of their findings, the authors associated PAGBC expression in GBC samples with clinical parameters: high expression of PAGBC correlated with an advanced tumor stage and a poor prognosis [87]. Additional in vitro studies showed that PAGBC promotes GBC carcinogenesis by enhancing cell proliferation, migration and invasion [87]. To explore the molecular oncogenic mechanism of PAGBC, the authors elaborated the potential of PAGBC to interact with miRs and predicted about 1000 potential miR binding sites for PAGBC, including binding sites for miR-133 and miR-511, which are described as tumor-suppressive miRs in gastric cancer and hepatocellular carcinoma, respectively [164]. An investigation regarding PAGBC and miRs-133 and 511 in BTC not only revealed direct binding of PAGBC to these miR species, but also inverse expression patterns between PAGBC and the two miRs, suggesting that PAGBC might act as a ceRNA in GBC [87].

To further unravel the oncogenic role of PAGBC via sponging miR-133 and miR-511, respectively, the authors performed target predictions for these two miR species and identified SOX4 as a potential target of miR-133b and PI3KR as a potential target of miR-511, respectively [87]. Knockdown of PAGBC in BTC cells not only resulted in enhanced levels of miR-133 and miR-511, but also in downregulation of SOX4 and PIK3R3, suggesting that PAGBC acts as an oncogenic ceRNA in GBC via the indirect regulation of these two oncogenic proteins [87]: SOX4 promotes carcinogenesis by transcriptional activation of several components of the PI3K/AKT [165], while PIK3R3 enhances progression of tumorigenesis through the AKT/mTOR pathway [166]. Lastly, the authors identified PABPC1 as a protein that directly interacts with PAGBC and that regulates the expression of PAGBC [87]. Interestingly, overexpression or knockdown of PAGBC had no significant effect on the expression levels of PABPC1, whereas knockdown of PABPC1 resulted in reduced PAGBC levels. Furthermore, PABPC1 showed a similar expression pattern to PAGBC in GBC patient samples, i.e., samples with a high expression of PAGBC also displayed high expression levels of PABPC1 [87].

## 8. H19

H19 was one of the first discovered lncRNAs [167]. Since then, numerous studies demonstrated the different mechanisms through which H19 contributes to tumorigenesis or may act as a tumor-suppressive lncRNA [168,169,170]. Regarding BTC, several studies demonstrated an oncogenic role of H19 including its association with unfavorable clinical characteristics such as advanced TNM-stage, tumor size and bad prognosis [65,66,67,68].

One major oncogenic mechanism of lncRNAs concerns their ability to sponge tumor-suppressive miRs [65,66,67]. Wang et al. showed that H19 is able to sponge the tumor-suppressive miR-194-5p [171] and that knockdown of H19 resulted in elevated miR-194-5p levels in BTC [66]. The oncogenic factor AKT2 was identified as a target of miR-194-5p in earlier studies [171]. Knockdown of H19 and concordant elevated miR-194-5p levels resulted in reduced AKT2 mRNA and protein levels in BTC cells, suggesting that H19 regulates AKT2 expression via sponging miR-194-5p [66].

Besides miR-194-5p, H19 also possesses the ability to sponge miR-342-3p, another established tumor-suppressive miR [67]. Again, knockdown of H19 resulted in enhanced levels of miR-342-3p and, in addition, reduced levels of the miR-342-3p target FOXM1, a transcriptions factor that was already demonstrated to possess oncogenic functions in BTC [67,172,173,174].

Previous studies indicate that oxidative stress can lead to inflammatory responses in cholangiocytes, thereby contributing to BTC development [175,176,177,178,179,180,181,182]. Interestingly, exposure of BTC cells to oxidative stress resulted in elevated levels of H19 in a let7a- and let-7b-dependent manner [65]. The authors demonstrated that by sponging the miRs let-7a and let-7b, H19 positively regulates the expression of the inflammation inducer IL-6 in BTC cells, which is especially interesting since IL-6 is described as an inducer of inflammatory processes and BTC development [65,183,184,185,186].

## 9. PVT1

One of the hallmarks of cancer is the reprogramming of glucose metabolism, better known as the Warburg effect, i.e., cancer cells prefer anaerobic glycolysis despite the availability of oxygen [187,188]. HK2 is a key enzyme in glucose metabolism and contributes to the Warburg effect, thus promoting tumorigenesis [90,189,190]. Chen et al. were able to connect the lncRNA PVT1 with HK2 and the Warburg effect in BTC [90]. They found that knockdown of PVT1 resulted in upregulation of miR-143, whereas overexpression of PVT1 reduced miR-143 levels. Moreover, they demonstrated a direct physical interaction between PVT1 and miR-143, suggesting that PVT1 acts as a ceRNA to sponge miR-143. As already shown in other cancer entities, HK2 is a target of miR-143 [191,192,193]. In BTC cells, knockdown of PVT1 resulted in significantly decreased HK2 mRNA and protein levels, whereas artificial overexpression of miR-143 reversed this effect [90]. These data suggest that PVT1 regulates HK2-levels via sponging miR-143 [90]. The authors underlined the significance of their findings and measured PVT1 levels in BTC patient specimens and observed high levels of PVT1 in tumor samples associated with an advanced TNM-stage, poorer overall survival and distant metastasis [90].

## 10. GBCDRlnc1

Chemoresistance of cancer cells is a complex and multifaceted process. Several mechanisms for the development of resistance are discussed and include enhanced expression of drug efflux pumps, altered metabolism, apoptosis resistance, angiogenesis, elevated DNA damage repair activity, and autophagy-induction [194,195,196]. Autophagy is an important process for the cells to eliminate dysfunctional and unnecessary cellular components, thereby supporting the cells to deal with stress situations such as infections and nutrition shortage [197,198]. Regarding cancer development, autophagy can be a double-edged sword. On the one hand, autophagy can have a tumor-suppressive role, as shown by the discovery of Beclin 1, an autophagy-related gene that is often deleted in cancer, resulting in diminished autophagy and enhanced tumorigenesis [199,200]. On the other hand, starvation-induced autophagy can be a crucial source of energy and nutrients for cancer cells, specifically in the case of metabolic stress [201,202]. Several studies demonstrate that stress tolerance, which is induced through cytoprotective autophagy, can contribute to resistance against chemotherapeutics [203,204]. Regulation of autophagy involves lncRNAs [64,205,206,207] which might also contribute to the development of chemoresistance [64]—an aspect that is especially interesting regarding BTC as BTC cells are often highly chemoresistant [64,208,209].

Cai et al. established doxorubicin-resistant GBC cells and compared the lncRNA expression profiles between the resistant cells and the non-resistant parental GBC cells [64]. Interestingly, they found several hundred lncRNA species with different expression profiles (about 450 lncRNAs were upregulated and about 260 lncRNAs were downregulated in the resistant cells) [64]. Of the highly significantly upregulated lncRNAs in the drug-resistant GBC cells, they identified a novel lncRNA termed GBCDRlnc1 [64]. To investigate its role in chemoresistance, they showed that GBCDRlnc1 induces resistance towards doxorubicin in an autophagy-dependent manner in BTC cells and that siRNA-based knockdown of GBCDRlnc1 sensitized cells towards doxorubicin treatment [64]. To further substantiate their findings, they measured GBCDRlnc1 expression in GBC patient tissue and adjacent non-tumor samples [64]. GBCDRlnc1 expression was significantly enhanced in tumor specimens and high expression of GBCDRlnc1 was associated with histological grade, TNM-stage and shorter survival [64]. To investigate the exact molecular mechanisms of GBCDRlnc1 in BTC cells, a knockdown was established in the doxorubicin-resistant cells [64]. Although knockdown of GBCDRlnc1 in these cells did not have any effects on cell proliferation and invasion, it affected the autophagy process: knockdown of GBCDRlnc1 resulted in elevated p62-levels and reduced the conversion of LC3-1 to LC3-II, a process which is suggested to happen in starvation-induced autophagy [64,210]. Moreover, PGK-1, an enzyme involved in glycolysis and activation of autophagy, was identified as a direct target of GBCDRlnc1 [64,211,212,213]. Treatment of GBC cells with siRNA against GBCDRlnc1 and a protein synthesis inhibitor resulted in a significantly shortened half-life of PGK-1 [64]. However, treatment of GBC cells with siRNA against GBCDRlnc1 and a proteasome inhibitor resulted in enhanced protein levels of PGK-1 [64]. Using co-immunoprecipitation, the authors further demonstrated that upon knockdown of GBCDRlnc1, ubiquitination of PGK-1 was significantly more frequent, suggesting that GBCDRlnc1 protects PGK-1 from ubiquitination and thereby positively regulates autophagy in BTC cells [64,212,213]. Interestingly, knockdown of GBCDRlnc1 enhanced the sensitivity of doxorubicin-resistant GBC cells towards doxorubicin, 5-fluorouracil and gemcitabine, underlining the involvement of this lncRNA species in chemoresistance [64].

## 11. NEAT1

Another oncogenic lncRNA that might also be involved in oncogenic processes in BTC and chemosensitivity is NEAT1 [86,214,215]. Zhang et al. observed overexpression of NEAT1 in BTC patient samples compared to adjacent non-tumor tissue [86]. On a molecular level, knockdown of NEAT1 resulted in decreased cell proliferation as well as diminished migration and invasion of BTC cells [85,86]. Furthermore, knockdown of NEAT1 resulted in enhanced levels of E-Cadherin accompanied by lower expression of mesenchymal markers Vimentin and N-Cadherin, suggesting that NEAT1 might be involved in regulation of EMT [86]. In fact, the authors showed that NEAT1 directly interacts with the negative E-Cadherin regulator EZH2 and that knockdown of NEAT1 resulted in reduction of EZH2 binding to the E-Cadherin promoter [86]. Besides EMT, there is also evidence that NEAT1 might be involved in chemosensitivity of BTC cells as knockdown of NEAT1 resulted in significantly higher sensitivity towards gemcitabine compared to control cells [85]. In this regard, the authors further found that BAP-1 (a chromatin modulator and potential tumor suppressor, [85,214,216]) directly regulates NEAT1 expression and that BTC cells and patient samples show inverse BAP-1/NEAT1 expression patterns [85]. BTC cells with low BAP-1 and high NEAT1 expression were less sensitive towards gemcitabine treatment, suggesting that the BAP-1/NEAT1-axis might be involved in the sensitivity of BTC cells towards the standard chemotherapeutic gemcitabine [85]. The authors also investigated the effect of differential BAP-1 expression and sensitivity of BTC cells towards the EZH2 inhibitor GSK126 [85,217]. They observed that low expression of BAP-1 (which correlated with high expression of NEAT1) resulted in lower GSK126 sensitivity of BTC cells [85]. Therefore, it will be interesting to see in future studies whether a direct connection between EZH2 and sensitivity towards EZH2 inhibition and NEAT1 expression exists.

## 12. SNHG1

The guidance of EZH2 to certain target promoters seems to be a frequently observed mechanism of lncRNAs in BTC (see Table 2). SNHG1 is upregulated in several cancer types, including non-small-cell lung carcinoma, prostate cancer, colon cancer and hepatocellular carcinoma [218,219,220,221]. Yu et al. demonstrated SNHG1 overexpression in BTC patient samples and that SNHG1 might act as an oncogenic lncRNA via its ability to guide EZH2 to target promoters for the silencing of tumor suppressor genes [93]. The authors performed an RNA transcriptome comparison between SNHG1 knockout and control cells and found differential expression of hundreds of transcripts, including the established tumor suppressor and EZH2 target CDKN1A, which showed significant upregulation upon SNHG1 knockout [93,222]. Using RNA and chromatin immunoprecipitation, the authors could not only demonstrate that SNHG1 can directly interact with EZH2, but also that knockdown of SNHG1 lowered EZH2 binding at the CDKN1A promoter, resulting in elevated protein levels of CDKN1A and suppression of proliferation of BTC cells [93].

Besides its role as a guidance-lncRNA for EZH2, SNHG1 also possesses the ability to sponge miR-140 in BTC cells as shown by Li et al. [94]. MiR-140 is a well-described tumor-suppressive miR [223,224,225]. In BTC cells, Li et al. identified TLR-4 as a downstream target of miR-140 [94]. TLR-4 is involved in activation of the NF-κB pathway, which is involved in the regulation of the innate and adaptive immunity and often found to be constitutively activated in several cancer types [226,227,228]. Knockdown of SNHG1 reduces sponging of miR-140, resulting in increased levels of miR-140 and consequently decreased TLR4 mRNA and protein levels as well as reduced NF-κB pathway activity. In contrast, overexpression of SNHG1 resulted in opposite effects, suggesting that SNHG1 can also act as an oncogenic ceRNA via sponging the tumor-suppressive miR-140 [94].

## 13. MALAT1

MALAT1, also known as NEAT2, is one of the most investigated oncogenic lncRNAs [229,230,231,232,233,234,235]. In BTC, MALAT1 possesses multifaceted oncogenic functions as it can act as a ceRNA, a guidance lncRNA for EZH2 and as a modulator of several oncogenic pathways, thereby directly modulating cell proliferation, metastasis, invasion and apoptosis [79,80,81,82,83,236]. In general, studies have shown that MALAT1 is upregulated in BTC and that elevated MALAT1 levels are associated with poor prognosis [79,80,81,82,83,236].

Up to now, two studies have investigated the ability of MALAT1 to act as a ceRNA in BTC and identified miR-363-3p and miR-206 as two MALAT1-mediated sponging targets [79,82]. Consequently, knockdown of MALAT1 resulted in elevated levels of miR-363-3p and miR-206 [79,82]. This effect was accompanied by enhanced expression of miR-363-3p target MCL-1 and miR-206 targets KRAS and ANXA2, which are known oncogenes and are overexpressed in BTC [79,82,237,238].

The activation of certain pathways through protein phosphorylation, such as PI3K/AKT or MEK/ERK is a well-described principle involved in carcinogenesis. Knockdown of MALAT1 resulted in reduced levels of phosphorylated PI3K, AKT, ERK, MEK, MAPK and JNK and overexpression of MALAT1 had the opposite effect [81,83]. Although the exact mechanisms remain unclear, these data strongly suggest a direct involvement of MALAT1 in the regulation of key oncogenic pathways in BTC.

Another oncogenic function of MALAT1 involves its direct interaction with EZH2: Lin et al. demonstrated that MALAT1 possesses the ability to guide EZH2 to the promoter region of the tumor suppressor ABI3BP for histone methylation and gene silencing [80,239]. Knockdown of MALAT1 as well as treatment of cells with the EZH2 inhibitor GSK343 resulted in reduced histone methylation at the ABI3BP promoter and, consequently, enhanced ABI3BP levels, which resulted in inhibition of BTC cell proliferation [80].

## 14. EZH2—A Major Target of LncRNA-based Regulation

As summarized in Table 2, both oncogenic, as well as tumor-suppressive lncRNA species are connected to EZH2 via different mechanisms. This observation is not only interesting, as EZH2 is overexpressed in multiple cancer types including BTC, but also since it is estimated that, in general, about 20% of all lncRNAs can interact with EZH2 [108,240,241,242]. EZH2 is the catalytic subunit of the PRC2 complex, an epigenetic regulator that specifically performs H3K27me3 using the methyl donor S-adenosyl methionine cofactor (SAM), thereby enabling the formation of heterochromatin and gene silencing [242]. Overexpression of EZH2 in cancer is associated with unfavorable clinical characteristics and shorter patient survival [240,243,244]. Interestingly, overexpression of EZH2 was also observed in BTC [108,245]. Using immunostaining, several studies could display an upregulated expression pattern of EZH2 in BTC patients linked with a poor overall survival, larger tumor size and lymph node metastasis compared to patients with no or a low EZH2 expression [108,246]. On a molecular level, p16 and PTEN, two well-known tumor suppressors, were shown to be negatively regulated by EZH2 [108,246,247]. Additionally, Yamaguchi et al. could demonstrate that EZH2 expression positively correlated with the expression of the proliferation marker KI-67 [108,248]. A better understanding of the interaction partners of EZH2 such as lncRNAs, could not only result in a better understanding of underlying mechanistic and regulatory mechanisms, but also in the conceptualization of new therapeutic approaches.

Current studies regarding BTC suggest that lncRNAs can interact with EZH2 in at least four different ways (see Figure 3). Recruitment of EZH2 and the PRC2 to its target genes is essential for epigenetic regulation [108,242]. As listed in Table 2, up to now, four lncRNAs (NEAT1, SNHG1, UCA1 and MALAT1) are described in BTC that are able to directly interact with EZH2 and thereby guide the PRC2 to its target gene loci for epigenetic gene silencing [80,86,93,101]. As described in the previous chapter, MALAT1 is able to inhibit the expression of the tumor suppressor ABI3BP gene in an EZH2-guidance-dependent manner, thereby promoting tumorigenesis in GBC [80], whereas NEAT1 and SNHG1 promote tumorigenesis via guiding EZH2 to the promoter region of the well-known tumor suppressors E-cadherin and p21, resulting in higher H3K27me3 levels at these gene loci [86,93]. Interestingly, the lncRNA UCA1 also guides EZH2 to the gene loci pf p21 and E-cadherin, causing epigenetic gene silencing of these tumor suppressors [101]. Knockdown of UCA1 resulted in reduced binding of EZH2 to target gene promoters and consequently lower H3K27me3 levels at their promoter regions [101]. Besides the ability of lncRNAs to guide EZH2 to specific loci, they also can act as specific assembly scaffolds in BTC cells [97]. Yi et al. demonstrated that the lncRNA SPRY4-IT1 is able to serve as a scaffold for assembling a regulatory complex consisting of EZH2, LSD1 and DNMT1, thereby enabling concerted binding of these epigenetic regulators to the same gene loci [97]. In BTC cells, this complex performs epigenetic silencing of the tumor suppressors KLF2 and LATS2 [97,154,249]. Knockdown of SPRY4-IT1 not only suppressed the binding of the complex to these specific gene loci, but also resulted in decreased H3K27me3- and increased H3K4me2 levels at these promoter regions and, consequently, enhanced expression of KLF2 and LATS2 [97].

Besides its ability to directly interact with EZH2 as a scaffold lncRNA, SPRY4-IT1 is also involved in regulation of EZH2 expression via its ability to sponge miR-101-3p, a negative regulator of EZH2 expression [97]. Sponging of miR-101-3p by SPRY4-IT1 resulted in increased protein levels of the EZH2 in BTC cells [97]. This observation is especially interesting, as (over)expression of certain ceRNA species might represent an important mechanism to regulate the protein levels of epigenetic regulators such as EZH2 in BTC [97]. A similar mechanism was published by Shou-Hua et al.: here, the authors demonstrated that the ceRNA MINCR sponges miR-26a-5p, another negative regulator of EZH2, which again resulted in upregulated EZH2-expression accompanied by enhanced proliferation of BTC cells [84].

Although most studies suggest an oncogenic role of lncRNAs in the context of EZH2 and BTC, certain lncRNA species might also act as tumor suppressors [107]. MEG3 causes ubiquitination and ultimately degradation of EZH2 via direct interaction, suggesting that MEG3 might act as a negative regulator of EZH2 [107]. This is especially interesting, as MEG3 was demonstrated to be downregulated in GBC tissue and as this downregulation was connected with unfavorable clinical characteristics and poor prognosis (see Table 1). This suggests that MEG3 might serve as a tumor-suppressive lncRNA in BTC via its ability to cause degradation of EZH2 [107]. These data not only underline the known importance of EZH2 in development and progression of BTC per se, but even more importantly, identify EZH2 as a nodal point for lncRNA-based oncogenic action in BTC.

## 15. Conclusions

Based on high-throughput technologies, more and more lncRNAs and their biological roles are being identified. LncRNAs act functionally in manifold ways, and increasing evidence shows that lncRNAs are deregulated in cancer. This insight not only adds a new layer of complexity to the understanding of tumor development and progression, but also demands the detailed investigation of lncRNAs as potential therapeutic targets and/or diagnostic markers. For example, the lncRNA PCA3, which is often found to be overexpressed in prostate cancer, is already used for clinical diagnostics, as PCA3 is detectable in urine samples from patients with prostate cancer, exemplifying the high potential of lncRNAs as tumor markers [250].

In BTC, most described lncRNAs act as oncogenic lncRNAs, although tumor-suppressive lncRNAs are also characterized. Interestingly, both oncogenic and tumor-suppressive lncRNAs, act via different mechanisms, e.g., as guide-lncRNAs, ceRNAs or scaffold-lncRNAs, which underlines the complexity of lncRNA-based regulation in BTC. Moreover, besides this categorization in functional archetypes, BTC-related lncRNAs also operate through specific interactions such as by ubiquitination of EZH2. Due to limited therapeutic options and dismal prognosis of BTC, a better understanding of underlying molecular processes in BTC development and progression is of utmost importance. LncRNAs harbor great potential as biomarkers and therapeutic targets. Currently, over ten clinical trials are ongoing, in which lncRNAs are evaluated as possible exosomal biomarkers in body fluids across several cancer entities (https://clinicaltrials.gov). Exosomes play a crucial role in cell-to cell communication as vehicles for mRNA, miRNA, lipids, proteins and lncRNAs [251]. In this regard, Li et al. could demonstrate an upregulation of the oncogenic lncRNA H19 in exosomes derived from cholangiocytes under pathological conditions in the bile acid, which demonstrates the possibility to use lncRNAs as biomarkers also in BTC [252]. A possible future therapeutic strategy for BTC might include the use of RNA interference substances against tumor-promoting lncRNAs or proteins that are essential for the lncRNA function [253,254]. The substance Patisiran represents a FDA-approved RNA-interfering treatment option for hereditary transthyretin amyloidosis and it will be interesting to see whether this concept can be translated into cancer research and RNA interfering drugs might be able to inhibit oncogenic lncRNAs [255]. On the contrary, the induction of tumor-suppressive lncRNAs might also represent a potential strategy against BTC [254]. As summarized in this review, lncRNAs are aberrantly expressed in BTC patient samples and are involved in diverse oncogenic processes in BTC. Therefore, future detailed investigation of lncRNAs in BTC might enhance our understanding of their molecular and functional roles and also allow for their utilization as tumor markers as well as the development of new therapeutic strategies.

## Figures and Tables

**Figure 1 jcm-09-01200-f001:**
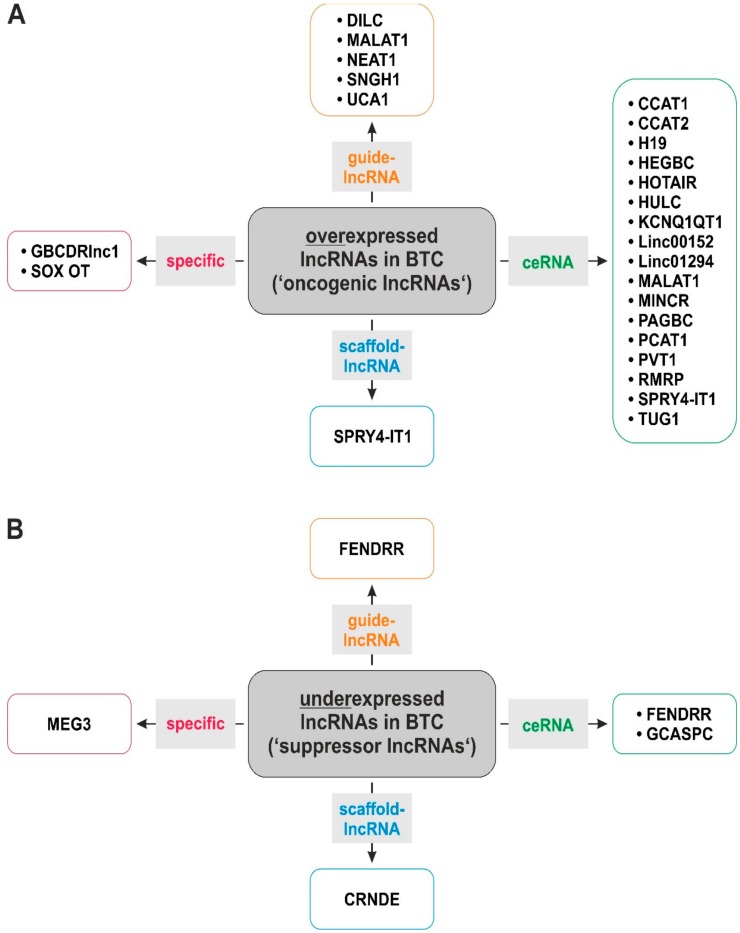
LncRNAs show functional diversity in biliary tract cancer as both oncogenic (**A**) and tumor-suppressive (**B**) lncRNAs. Abbreviations: lncRNA: long-noncoding RNA.

**Figure 2 jcm-09-01200-f002:**
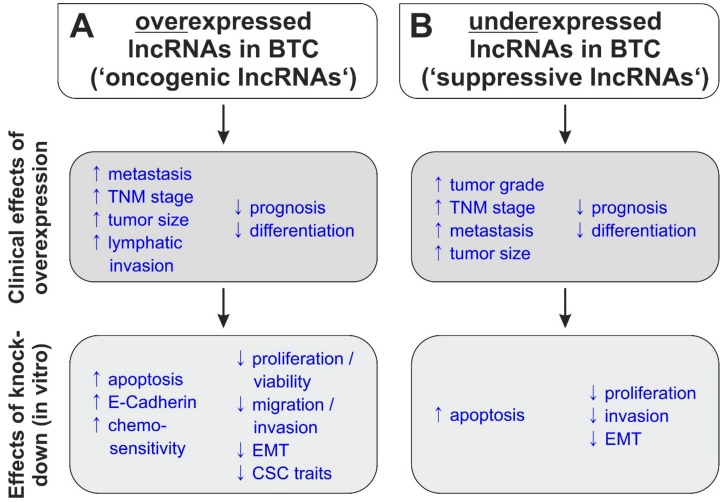
(**A**) Overexpression of oncogenic lncRNAs in BTC patient specimens is associated with unfavorable clinical characteristics. Knockdown of oncogenic lncRNAs in BTC in vitro models results in reduction of oncogenic characteristics and initiation of apoptosis, E-Cadherin expression and enhanced chemosensitivity. (**B**) Under-expression of suppressive lncRNAs in BTC patient specimens is associated with unfavorable clinical characteristics. Artificial overexpression of suppressive lncRNAs in BTC in vitro models reduced BTC cell proliferation, invasion and EMT characteristics and induced apoptosis. Abbreviations: BTC: biliary tract cancer; EMT: epithelial-to-mesenchymal transition; lncRNA: long-noncoding RNA.

**Figure 3 jcm-09-01200-f003:**
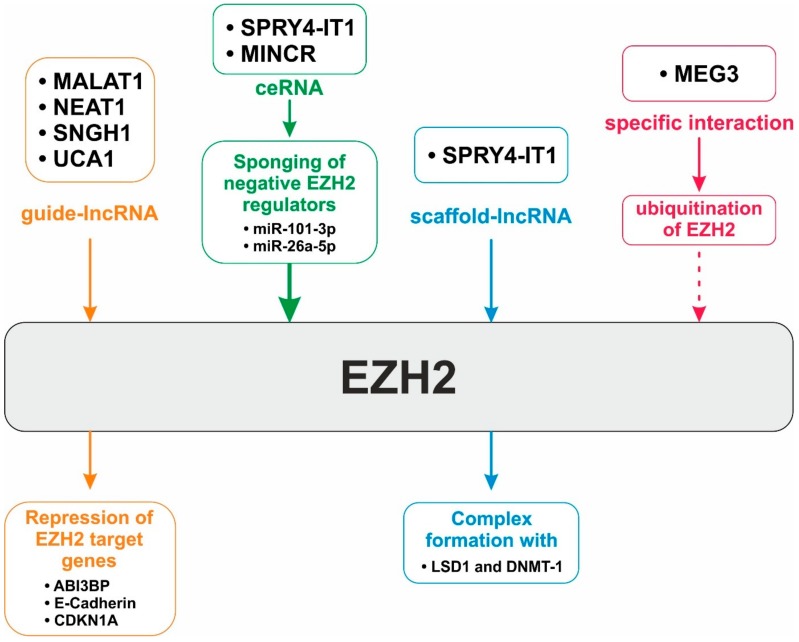
Overview of the versatility of lncRNA-based regulation of EZH2 in biliary tract cancer. Guide-lncRNAs directly interact with EZH2 and guide EZH2 (as enzymatic part of the polycomb repressive complex 2) to specific target gene promoters for epigenetic gene silencing. Via sponging negative EZH2 regulators (miR-101-3p and miR-26a-5p), ceRNAs positively regulate EZH2 protein levels. SPRY4-IT1 serves as a scaffold lncRNA to allow the assembly and thereby the functionality of an epigenetic regulatory complex consisting of EZH2, LSD1 and DNMT1. MEG3 negatively regulates EZH2 protein expression via initiation of EZH2 ubiquitination and, consequently, EZH2 degradation. Abbreviations: lncRNA: long-noncoding RNA; miR: micro-RNA.

**Table 1 jcm-09-01200-t001:** Overview of up-regulated (‘oncogenic‘) and down-regulated (‘suppressive‘) lncRNAs in biliary tract cancer patient specimens compared to healthy tissue and the associated clinical effects.

	LncRNA	Tissue	Clinical Effects						Ref
			Prognosis	Tumor Size	TNM Status	Differentiation	Metastasis	Lymphatic Invasion	
**Up-Regulated**	AFAP-AS1	CCA/GBC	↓	↑	↑				[53,54]
	ANRIL	IHC	↓						[55,56]
	CPS1-IT1	IHC	↓						[57]
	CCAT1	IHC	↓		↑	↓		↑	[58,59,60]
	CCAT2	IHC	↓		↑		↑		[61]
	DILC	GBC	*						[62]
	EPIC1	CCA	*						[63]
	GBCDRlnc1	GBC			↑				[64]
	H19	GBC	↓	↑	↑				[65,66,67,68]
	HEGBC	GBC	↓						[69,70]
	HOTAIR	CCA	↓	↑	↑				[71,72]
	HOXA-AS2	GBC		↑	↑				[73]
	HULC	CCA	*						[65]
	KCNQ1QT1	CCA	↓						[74]
	Linc00152	GBC	↓		↑			↑	[75,76]
	Linc01296	CCA	↓				↑		[77]
	Loc344887	GBC		↑					[78]
	MALAT1	GBC	↓	↑			↑		[79,80,81,82,83]
	MINCR	GBC	↓	↑			↑		[84]
	NEAT1	CCA	*						[85,86]
	PAGBC	GBC		↑			↑		[87]
	PANDAR	CCA	↓		↑			↑	[88]
	PCAT1	EHC	*						[89]
	PVT1	GBC	↓		↑		↑		[90]
	RMRP	CCA	*						[91]
	ROR	GBC	↓	↑			↑		[92]
	SNHG1	CCA	*						[93,94]
	SOX OT2	CCA	↓		↑			↑	[95,96]
	SPRY4-IT1	CCA/GBC	↓		↑				[97,98]
	TP73-AS1	CCA		↑	↑				[99]
	TUG1	GBC					↑		[100]
	UCA1	GBC	↓	↑	↑		↑		[101]
**Down-** **Regulated**	CRNDE	GBC	*						[102]
	FENDRR	CCA	*						[103,104]
	GCASPC	GBC	↓	↑					[105]
	LET1	GBC	↓			↓	↑		[106]
	MEG3	GBC	↓		↑		↑		[107]

Abbreviations: CCA: cholangiocellular carcinoma; EHC: extrahepatic cholangiocarcinoma; GBC: gallbladder cancer; IHC: intrahepatic cholangiocarcinoma; lncRNA: long-noncoding RNA. * clinical effects not described.

**Table 2 jcm-09-01200-t002:** Overview of the molecular mechanism predicted targets and effects of knockdown of oncogenic and suppressive lncRNAs based on in vitro studies in biliary tract cancer.

**oncogenic lncRNAs**	**lncRNA**	**Molecular Mechanism**					**Predicted Targets**	**Effects of Knockdown**									**Ref**
								**↓**						**↑**			
		ceRNA	scaffold	guide lncRNA	specific	not described		proliferation	cell viability	migration	invasion	EMT	CSC traits	chemosensitivity	apoptosis	E-Cadherin	
	AFAP-AS1					x		x		x	x	x					[53,54]
	ANRIL					x		x			x				x		[55,56]
	CPS1-IT1					x		x							x		[57]
	CCAT1	x					miR-152, miR-218-5p			x	x	x					[58,59,60]
	CCAT2	x						x		x	x		x				[61]
	DILC			x				x					x				[62]
	EPIC1					x		x					x		x		[63]
	GBCDRlnc1				x		PGK1							x			[64]
	H19	x					miR-342-2p, miR-195-5p, let7a/b	x		x	x					x	[65,66,67,68]
	HEGBC	x					miR-502-3p		x	x					x		[69,70]
	HOTAIR	x					miR-130a	x			x	x			x	x	[71,72]
	HOXA-AS2					x		x		x	x	x				x	[73]
	HULC	x					miR-372, miR-373			x	x						[65]
	KCNQ1QT1	x					miR-140-5p				x	x			x		[74]
	Linc00152	x					miR-138	x			x	x			x	x	[75,76]
	Linc01296	x					miR-5095		x	x	x				x		[77]
**oncogenic lncRNAs (continued)**	**lncRNA**	**Molecular Mechanism**					**Predicted Targets**	**Effects of Knockdown**									**Ref**
								**↓**						**↑**			
		ceRNA	scaffold	guide lncRNA	specific	not described		proliferation	cell viability	migration	invasion	EMT	CSC traits	chemosensitivity	apoptosis	E-Cadherin	
	Loc344887					x		x		x		x				x	[78]
	MALAT1	x		x			miR-206, miR-363-3p/EZH2	x		x	x	x			x	x	[79,80,81,82,83]
	MINCR	x					miR-26a-5p	x		x	x				x		[84]
	NEAT1			x			EZH2	x		x	x		x			x	[85,86]
	PAGBC	x					miR-133, miR-511			x	x						[87]
	PANDAR					x		x		x	x	x			x	x	[88]
	PCAT1	x					miR-122	x		x	x				x		[89]
	PVT1	x					miR-143	x		x	x						[90]
	RMRP	x					miR-217		x						x		[91]
	ROR					x		x		x	x	x					[92]
	SNHG1			x			EZH2	x		x	x				x		[93,94]
	SOX2 OT				x			x									[95,96]
	SPRY4-IT1	x	x				miR-101-3p; EZH2/LSD1/DNMT1	x		x	x				x	x	[97,98]
	TP73-AS1					x		x							x		[99]
	TUG1	x					miR-300	x				x					[100]
	UCA1			x			EZH2	x									[101]
**tumor-suppressive lncRNAs**	**lncRNA**	**Molecular Mechanism**					**Predicted Targets**	**Effects of Overexpression**									**Ref**
								**↓**							**↑**		
		ceRNA	scaffold	guide lncRNA	specific	not described		proliferation	invasion	EMT	specific	-	-	-	apoptosis	-	
	CRNDE		x				DMBT/c-IAP1				x						[102]
	FENDRR	x		x			miR-18b-5p; Survivin				x						[103,104]
	GCASPC	x					miR-17-3p	x									[105]
	LET1					x			x						x		[106]
	MEG3				x		EZH2	x	x	x					x		[107]

Abbreviations: CSC: cancer stem cells; EMT: epithelial-to-mesenchymal-transition; lncRNA: long-noncoding RNA; miR: micro-RNA.

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
