# Peer review of "Long Non-Coding RNAs in Biliary Tract Cancer—An Up-to-Date Review"

_jcm, 2020, doi:10.3390/jcm9041200_

Round 1

Reviewer 1 Report

The manuscript was well written, which introduced and presented the roles of lncRNAs in biological processes in the BTC with clinical implication, logically and comprehensively.

1. As most paradigms of lncRNAs described in the manuscript function as ceRNA, it will be better if the authors could highlight and explain more about the ceRNA mechanism or networks in the "long non-coding RNAs" section.  

2. EZH2 has been illustrated as an essential target for lncRNAs in BTC. I recommend the authors to introduce more about the roles of EZH2 or EZH2-associated pathways in the hallmarks of BTC, which will be more convincing that studying lncRNAs targeting EZH2 networks could contribute to pushing the efficient therapy for BTC forward. 

3. Improve the quality of Figures 1 & 2; some words were blurry. 

I recommend publishing this review after minor revisions. 

Author Response

Reviewer 1:

The manuscript was well written, which introduced and presented the roles of lncRNAs in biological processes in the BTC with clinical implication, logically and comprehensively.

We thank the reviewer for his/her positive comment on our manuscript.

As most paradigms of lncRNAs described in the manuscript function as ceRNA, it will be better if the authors could highlight and explain more about the ceRNA mechanism or networks in the "long non-coding RNAs" section.

As suggested, we have added a paragraph, in which we briefly discuss the importance and complexity of ceRNA-based transcriptional regulatory networks in post-transcriptional gene expression regulation. In addition, we have added information of a study that describes a ceRNA regulatory network in BTC.

EZH2 has been illustrated as an essential target for lncRNAs in BTC. I recommend the authors to introduce more about the roles of EZH2 or EZH2-associated pathways in the hallmarks of BTC, which will be more convincing that studying lncRNAs targeting EZH2 networks could contribute to pushing the efficient therapy for BTC forward.

As suggested, we have extended the section regarding EZH2 and BTC and added information about EZH2 expression and BTC outcome as well as molecular consequences of EZH2 overexpression in BTC.

Improve the quality of Figures 1 & 2; some words were blurry.

We checked the original figures and did not recognize any quality problems. However, we found that inclusion of the figures in the manuscript file resulted in quality problems regarding the frames of the boxes. We apologize for this mistake and modified the figures accordingly.

I recommend publishing this review after minor revisions.

Reviewer 2 Report

In the present review, the Authors aimed to summarize the roles and the findings regarding the term long non-coding RNA (lncRNA) in biliary tract cancer (BTC), giving a view of the molecular and clinical involvements of aberrant lncRNA expression, together with the regulatory functions of some lncRNA linked to BTC. It is a well-written article with important and detailed information. In my opinion, the Authors should enrich only some points:

  • In the first part of the Introduction section, they have divided BTC into intrahepatic (IHC), extrahepatic (EHC) and gallbladder cancer (GBC). However, inside the intrahepatic group, additional subdivisions exist, such as the mixed and mucinous subtypes, correlated to the histological aspects. But also, according to the anatomical location we can define them intrahepatic, perihilar or distal (PMID:28994423). All these aspects are important since they are linked to the epidemiology, pathogenesis and management of cholangiocarcinoma.
  • In the DILC section, the Authors have introduced the cancer stem cell (CSC) model, could they explain better their specific characteristics?

Author Response

Reviewer 2:

In the present review, the Authors aimed to summarize the roles and the findings regarding the term long non-coding RNA (lncRNA) in biliary tract cancer (BTC), giving a view of the molecular and clinical involvements of aberrant lncRNA expression, together with the regulatory functions of some lncRNA linked to BTC. It is a well-written article with important and detailed information. In my opinion, the Authors should enrich only some points:

We thank the reviewer for his/her positive comment on our manuscript.

In the first part of the Introduction section, they have divided BTC into intrahepatic (IHC), extrahepatic (EHC) and gallbladder cancer (GBC). However, inside the intrahepatic group, additional subdivisions exist, such as the mixed and mucinous subtypes, correlated to the histological aspects. But also, according to the anatomical location we can define them intrahepatic, perihilar or distal (PMID:28994423). All these aspects are important since they are linked to the epidemiology, pathogenesis and management of cholangiocarcinoma.

As suggested, we have added additional information regarding the anatomic and histopathological classifications of BTC in the introduction.

In the DILC section, the Authors have introduced the cancer stem cell (CSC) model, could they explain better their specific characteristics?

As suggested, we have extended the section regarding CSCs and have added information regarding CSC models, CSC characteristics and CSCs in BTC.